# DeepMoCap: Deep Optical Motion Capture Using Multiple Depth Sensors and Retro-Reflectors

**DOI:** 10.3390/s19020282

**Published:** 2019-01-11

**Authors:** Anargyros Chatzitofis, Dimitrios Zarpalas, Stefanos Kollias, Petros Daras

**Affiliations:** 1Centre for Research and Technology Hellas, Information Technologies Institute, 6th km Charilaou-Thermi, 57001 Thermi, Thessaloniki, Greece; zarpalas@iti.gr (D.Z.); daras@iti.gr (P.D.); 2National Technical University of Athens, School of Electrical and Computer Engineering, Zografou Campus, Iroon Polytechniou 9, 15780 Zografou, Athens, Greece; stefanos@cs.ntua.gr; 3School of Computer Science, University of Lincoln, Brayford LN67TS, UK

**Keywords:** motion capture, deep learning, retro-reflectors, retro-reflective markers, multiple depth sensors, low-cost, deep mocap, depth data, 3D data, 3D vision, optical mocap, marker-based mocap

## Abstract

In this paper, a marker-based, single-person optical motion capture method (DeepMoCap) is proposed using multiple spatio-temporally aligned infrared-depth sensors and retro-reflective straps and patches (reflectors). DeepMoCap explores motion capture by automatically localizing and labeling reflectors on depth images and, subsequently, on 3D space. Introducing a non-parametric representation to encode the temporal correlation among pairs of colorized depthmaps and 3D optical flow frames, a multi-stage Fully Convolutional Network (FCN) architecture is proposed to jointly learn reflector locations and their temporal dependency among sequential frames. The extracted reflector 2D locations are spatially mapped in 3D space, resulting in robust 3D optical data extraction. The subject’s motion is efficiently captured by applying a template-based fitting technique on the extracted optical data. Two datasets have been created and made publicly available for evaluation purposes; one comprising multi-view depth and 3D optical flow annotated images (DMC2.5D), and a second, consisting of spatio-temporally aligned multi-view depth images along with skeleton, inertial and ground truth MoCap data (DMC3D). The FCN model outperforms its competitors on the DMC2.5D dataset using 2D Percentage of Correct Keypoints (PCK) metric, while the motion capture outcome is evaluated against RGB-D and inertial data fusion approaches on DMC3D, outperforming the next best method by 4.5% in total 3D PCK accuracy.

## 1. Introduction

Human pose tracking, also known as motion capture (MoCap), has been studied for decades and is still a very active and challenging research topic. MoCap is widely used in industries such as gaming, virtual/augmented reality, film making and computer graphics animation, among others, as a means to provide body (and/or facial) motion data for virtual character animation, humanoid robot motion control, computer interaction, and more. To date, a specialized computer vision and marker-based MoCap technique, called Optical Motion Capture [1], constitutes the gold-standard for accurate and robust motion capture [2]. Optical MoCap solutions [3,4,5] employ multiple optical sensors and passive or active markers (passive markers are coated with retro-reflective material to reflect light, while active markers are powered to emit it; for passive marker-based MoCap systems, IR emitters are also used to cast IR light on the markers) placed on the body of the subject to be captured. The 3D positions of the markers are extracted by intersecting the projections of two or more spatio-temporally aligned optical sensors. These solutions precisely capture the body movements, i.e., the body joint 3D positions and orientations per frame in high frequency (ranging from 100 to 480 Hz).

In the last decade, professional optical MoCap technologies have seen rapid development due to the high demand of the industry and the strong presence of powerful game engines [6,7,8], allowing for immediate and easy consumption of motion capture data. However, difficulties in using traditional optical MoCap solutions still exist. Purchasing a professional optical MoCap system is extremely expensive, while the equipment is cumbersome and sensitive. With respect to its setup, several steps should be carefully followed, ideally by a technical expert, to appropriately setup the required hardware/software and to rigidly install the optical MoCap cameras on walls or other static objects [1,9]. In addition, time-expensive and non-trivial post-processing is required for optical data cleaning and MoCap data production [10]. To this end, there still exists an imperative need for robust MoCap methods that overcome the aforementioned barriers.

In this paper, a low-cost, fast motion capture method is proposed, namely DeepMoCap, approaching marker-based optical MoCap by combining Infrared-Depth (IR-D) imaging, retro-reflective materials (similarly to passive markers usage) and fully convolutional neural networks (FCN). In particular, DeepMoCap deals with single-person, marker-based motion capture using a set of retro-reflective straps and patches (reflector-set: a set of retro-reflective straps and patches, called reflectors for the sake of simplicity) from off-the-shelf materials (retro-reflective tape) and relying on the feed of multiple spatio-temporally aligned IR-D sensors. Placing reflectors on and IR-D sensors around the subject, the body movements are fully captured, overcoming one-side view limitations such as partial occlusions or corrupted image data. The rationale behind using reflectors is the exploitation of the intense reflections they provoke to the IR streams of the IR-D sensors [11,12], enabling their detection on the depth images. FCN [13], instead of using computationally expensive fully connected layers, are applied on the multi-view IR-D captured data, resulting in reflector 2D localization and labeling. Spatially mapping and aligning the detected 2D points to 3D Cartesian coordinates with the use of depth data and intrinsic and extrinsic IR-D camera parameters, enables frame-based 3D optical data extraction. Finally, the subject’s motion is captured by fitting an articulated template model to the sequentially extracted 3D optical data.

The main contributions of the proposed method are summarized as follows:A low-cost, robust and fast optical motion capture framework is introduced, using multiple IR-D sensors and retro-reflectors. Contrary to the gold-standard marker-based solutions, the proposed setup is flexible and simple, the required equipment is low-cost, the 3D optical data are automatically labeled and the motion capture is immediate, without the need for post-processing.To the best of our knowledge, DeepMoCap is the first approach that employs fully convolutional neural networks for automatic 3D optical data localization and identification based on IR-D imaging. This process, denoted as “2D reflector-set estimation”, replaces the manual marker labeling and tracking tasks required in traditional optical MoCap.The convolutional pose machines (CPM) architecture proposed in [14] has been extended, inserting the notion of time by adding a second 3D optical flow input stream and using 2D Vector Fields [14] in a temporal manner.A pair of datasets consisting of (i) multi-view colorized depth and 3D optical flow annotated images and (ii) spatio-temporally aligned multi-view depthmaps along with Kinect skeleton, inertial and ground truth MoCap [5] data, have been created and made publicly available (https://vcl.iti.gr/deepmocap/dataset).

The remainder of this paper is organized as follows: Section 2 overviews related work; Section 3 explains in detail the proposed method for 2D reflector-set estimation and motion capture; Section 4 presents the published datasets; Section 5 gives and describes the experimental frameworks and results; finally, Section 6 concludes the paper and discusses future work.

## 2. Related Work

The motion capture research field consists of a large variety of research approaches. These approaches are marker-based or marker-less, while the input they are applied on is acquired using RGB or RGB-D IR/stereo cameras, optical motion capture or inertial among other sensors. Moreover, they result in single- or multi-person 2D/3D motion data outcome, performing in real-time, close to real-time or offline. The large variance of the MoCap methods resulting from the multiple potential combinations of the above led us to classify and discuss them according to an “Input–Output” perspective. At first, approaches that consume 2D data and yield 2D and 3D motion capture outcome are presented. These methods are highly relevant to the present work since, in a similar fashion, the proposed FCN approaches 2D reflector-set estimation by predicting heat maps for each reflector location and optical flow. Subsequently, 3D motion capture methods that acquire and process 2.5D or 3D data from multiple RGB-D cameras similarly to the proposed setup are discussed. Finally, methods that fuse RGB-D with inertial data for 3D motion capture are presented, including one of the methods that are compared against DeepMoCap in the experimental evaluation.

**2D Input–2D Output:** Intense research effort has been devoted to the 2D pose recovery task for MoCap, providing efficient methods being effective in challenging and “in-the-wild” datasets. Pose machines architectures for efficient articulated pose estimation [15] were recently introduced, employing implicit learning of long-range dependencies between image and multi-part cues. Later on, multi-stage pose machines were extended to CPM [14,16,17] by combining pose machine rationale and FCN, allowing for learning feature representations for both image and spatial context directly from image data. At each stage, the input image feature maps and the outcome given by the previous stage, i.e., confidence maps and 2D vector fields, are used as input refining the predictions over successive stages with intermediate supervision. Beyond discrete 2D pose recovery, 2D pose tracking approaches have been introduced imposing the sequential geometric consistency by capturing the temporal correlation among frames and handling severe image quality degradation (e.g., motion blur or occlusions). In [18], the authors extend CPM by incorporating a spatio-temporal relation model and proposing a new deep structured architecture. Allowing for end-to-end training of body part regressors and spatio-temporal relation models in a unified framework, this model improves generalization capabilities by spatio-temporally regularizing the learning process. Moreover, the optical flow computed for sequential frames is taken into account by introducing a flow warping layer that temporally propagates joint prediction heat maps. Luo et al. [19] also extend CPM to capture the spatio-temporal relation between sequential frames. The multi-stage FCN of CPM has been re-written as a Recurrent Neural Network (RNN), also adopting Long Short-Term Memory (LSTM) units between sequential frames, to effectively learn the temporal dependencies. This architecture, called LSTM Pose Machines, captures the geometric relations of the joints in time, increasing motion capture stability.

**2D Input–3D Output:** During the last years, computer vision researchers approach 3D pose recovery on single-view RGB data [20,21,22,23] for 3D MoCap. In [24], a MoCap framework is introduced, realizing 3D pose recovery, that consists of a synthesis between discriminative image-based and 3D pose reconstruction approaches. This framework combines image-based 2D part location estimates and model-based 3D pose reconstruction, so that they can benefit from each other. Furthermore, to improve the robustness of the approach against person detection errors, occlusions, and reconstruction ambiguities, temporal filtering is imposed on the 3D MoCap task. Similarly to CPM, 2D keypoint confidence maps representing the positional uncertainty are generated with a FCN. The generated maps are combined with a sparse model of 3D human pose within an Expectation-Maximization framework to recover the 3D motion data. In [25], a real-time method that estimates temporally consistent global 3D pose for MoCap from one single-view RGB video is presented, extending top performing single-view RGB convolutional neural network (CNN) methods for MoCap [20,23]. For best quality at real-time frame rates, a shallower variant is extended to a novel fully convolutional formulation, enabling higher accuracy in 2D and 3D pose regression. Moreover, CNN-based joint position regression is combined with an efficient optimization step for 3D skeleton fitting in a temporally stable way, yielding MoCap data. In [26], an existing “in-the-wild” dataset of images with 2D pose annotations is augmented by applying image-based synthesis and 3D MoCap data. To this end, a new synthetic dataset with a large number of new “in-the-wild” images is created, providing the corresponding 3D pose annotations. On top of that, the synthetic dataset is used to train an end-to-end CNN architecture for motion capture. The proposed CNN clusters a 3D pose to *K* pose classes on per-frame basis, while a K-way CNN classifier returns a distribution over probable pose classes.

**2.5D / 3D Input–3D Output:** With respect to 2.5D/3D data acquisition and 3D MoCap, particular reference should be made to the Microsoft Kinect sensor [27], beyond its discontinuation, since it was the first low-cost RGB-D sensor for depth estimation and 3D motion capture, leading to a massive release of MoCap approaches that use the Kinect streams or are compared to Kinect motion capture [28,29,30,31]. This sensor triggered the massive production of low-cost RGB-D cameras, allowing a wide community of researchers to study on RGB-D imaging and, subsequently, resulting in a plethora of efficient MoCap approaches applied on 2.5D/3D data [32,33,34,35,36]. In [37], a multi-view and real-time method for multi-person motion capture is presented. Similarly to the proposed setup, multiple spatially aligned RGB-D cameras are placed to the scene. Multi-person motion capture is achieved by fusing single-view 2D pose estimates from CPM, as proposed in [14,16], extending them to 3D by means of depth information. Shafaei et al. [38] use multiple externally calibrated RGB-D cameras for 3D MoCap, splitting the multi-view pose estimation task into (i) dense classification, (ii) view aggregation, and (iii) pose estimation steps. Applying recent image segmentation techniques to depth data and using curriculum learning, a CNN is trained on purely synthetic data. The body parts are accurately localized without requiring an explicit shape model or any other a priori knowledge. The body joint locations are then recovered by combining evidence from multiple views in real-time, treating the problem of pose estimation for MoCap as a linear regression. In [39], a template-based fitting to point-cloud motion capture method is proposed using multiple depth cameras to capture the full body motion data, overcoming self-occlusions. A skeleton model consisting of articulated ellipsoids equipped with spherical harmonics encoded displacement and normal functions is used to estimate the 3D pose of the subject.

**Inertial (+2.5D) Input–3D Output:** Inertial data [40,41,42,43], as well as their fusion with 2.5D data from RGB-D cameras, are also used to capture the human motion. In [44], Kinect for Xbox One [27] skeleton tracking is fused with inertial data for motion capture. In particular, inertial sensors are placed on the limbs and the torso of the subject to provide body bone rotational information by applying orientation filtering on inertial data. Initially, using Kinect, the lengths of the bones and the rotational offset between the Kinect and inertial sensors coordinate systems are estimated. Then, the bones hierarchically follow the inertial sensor rotational movements, while the Kinect camera provides the root 3D position. In a similar vain, a light-weight, robust method [45] for real time motion and performance capture is introduced using one single depth camera and inertial sensors. Considering that body movements follow articulated structures, this approach captures the motion by constructing an energy function to constrain the orientations of the bones using the orientation measurements of their corresponding inertial sensors. In [46], inertial motion capture is achieved on the basis of a very sparse inertial sensor setup, i.e., two units placed on the wrists and one on the lower trunk, and ground contact information. Detecting and identifying ground contact from the lower trunk sensor signals and combining this information with a fast database look-up enables data-driven motion reconstruction.

Despite the appearance of the aforementioned methods, traditional marker-based optical MoCap still remains the top option for robust and efficient motion capture. That is due to the stability of the marker-based optical data extraction and the deterministic way of motion tracking. To this end, the proposed method approaches marker-based optical motion capture, however overcoming restrictions of traditional marker-based optical MoCap solutions by:using off-the-shelf retro-reflective straps and patches to replace the spherical retro-reflective markers, which are sensitive due to potential falling off;automatically localizing and labeling the reflectors on a per-frame basis without the need for manual marker labeling and tracking;extracting the 3D optical data by means of the IR-D sensor depth.

Taking the above into consideration, DeepMoCap constitutes an alternative, low-cost and flexible marker-based optical motion capture method that results in high quality and robust motion capture outcome.

## 3. Proposed Motion Capture Method

DeepMoCap constitutes an online, close to real-time marker-based approach that consumes multi-view IR-D data and results in single-person 3D motion capture. The pipeline of the proposed method, depicted in Figure 1, is summarized as follows: A set of retro-reflective straps and patches is placed on the subject’s body.Placing multiple calibrated and synchronized IR-D sensors around the subject to fully capture the body movements, IR-D raw data are acquired and processed, giving multi-view pairs of colorized depth and 3D optical flow.Each pair is fed to a FCN model, resulting in 2D reflector-set estimation per view.The reflector-set estimates are spatially mapped in 3D space using the depth data and the intrinsic and extrinsic calibration camera parameters. The resulting 3D point sets are fused, resulting in 3D optical data outcome.A template-based articulated structure is registered and fitted to the subject’s body. 3D motion capture is achieved by applying forward kinematics to this structure based on the extracted 3D optical data.

The pipeline steps are accordingly described in the following sections.

### 3.1. Reflector-Set Placement

The reflector-set placement has been designed to provide robust and highly informative motion capture data, i.e., capturing large number of degrees of freedom (DoFs). The selected placement is shown in Figure 2, consisting of a set of 26 reflectors $R_{i}\in \left\{R_{1},\ldots ,R_{26}\right\}$, 16 patches and 10 straps, enabling motion capture by fitting an articulated body structure of 40 DoFs. The use of both straps and patches has been chosen due to the fact that the straps are 360°-visible on cylindrical body parts (i.e., limbs), while patches have been used on the body parts where strap placement is not feasible, i.e., torso, head and hands. Aiming to highlight the distinction between the front and the back side of the body, the reflective patches are not symmetrically placed. On the front side, two reflector patches are placed on the head, two on the chest and one on the spine middle, while on the back side, one is placed on the head, one on the back and one on the spine middle. The retro-reflective material used to create the reflector-set is the off-the-shelf 2-inch reflective tape used in protective clothing [47]. Following carefully the matching between the reflectors and the body-parts of the subject as depicted in Figure 2, the reflector-set placement is a fast procedure (it lasts approximately 2 min), since the sticky straps and patches are effortlessly placed, not requiring high placement precision (it is enough to be approximately placed to the body part locations shown in Figure 2).

### 3.2. Raw Data Acquisition and Processing

After the reflector-set placement, the subject is ready to be captured. Let us consider the use of *N* IR-D devices, thus, *N* is also the number of views v∈{1,…,N}. Using the multi-Kinect for Xbox One capturing setup proposed in [48], spatio-temporally aligned multi-view IR-D data are acquired. All reflector regions have distinguishable pixel values on IR images $I_{IR}^{v}$ (Figure 3a), thus, applying binary hard-thresholding, the binary mask $I_{IR_{m}}^{v}$ of the reflectors is extracted (Figure 3b). The corresponding regions on the raw depth images $I_{D}^{v}$ (Figure 3c) have zero values due to the retro-reflections.

The multi-view IR-D raw data are processed before feeding the FCN. On the one hand, the IR-D frames are jointly processed in order to compute the 3D optical flow $I_{F}^{v}$ of the body movements. In the present work, the primal-dual algorithm proposed in [49] is considered due to its demonstrated efficiency on relevant computer vision tasks, such as interaction-based RGB-D object recognition [50] and 3D action recognition [51]. In particular, the 3D motion vectors between two pairs of IR-D images and their magnitude are computed. The 3D flow and its magnitude are then colorized by normalizing each axis values and transforming the 3D motion vectors into a three-channel image. On the other hand, the depth images $I_{D}^{v}$ are colorized applying JET color map conversion. Finally, the reflector mask $I_{IR_{m}}^{v}$ is subtracted from the colorized depth images, resulting in colorized depth with reflector black regions, $I_{CD}^{v}$, facilitating the detection of the reflectors. The colorization step for both streams is required in order to allow the usage of the proposed FCN, initialized by the first 10 layers of VGG-19 [52]. An example of the processed multi-view outcome is shown in Figure 4.

### 3.3. 2D Reflector-Set Estimation Using FCN

The major challenge of the proposed method is the efficient localization and identification task of the reflectors placed on the subject’s body. Studying the recent literature in 2D localization on RGB images, the efficiency of deep neural networks in complex tasks such as articulated 2D pose estimation is remarkable and, therefore, considered appropriate for the present challenge. To this end, a deep learning approach is introduced extending the multi-stage CPM architecture in order to localize and identify the reflectors on the body. In particular, a multi-stage fully convolutional network is trained to directly operate on intermediate confidence maps and optical flow 2D vector fields, instead of Part Affinity Fields (PAF) between 2D keypoints, implicitly learning image-dependent spatial models of the reflectors locations among sequential frames.

Despite the similarities between 2D pose and reflector-set estimation, both being solvers of 2D localization problems on color images, there exist noteworthy differences. Cao et al. [14] efficiently address the problem of 2D pose estimation in large, “in-the-wild” RGB datasets [53], resulting in accurate estimates in a variety of data showing multiple people in different environmental and lightning conditions. In contrast, the reflector-set estimation is applied in more “controlled” conditions; (i) the input depth data lie within a narrow range, (ii) the reflector regions have clearly distinguishable pixel values on IR images, (iii) there is only one subject to be captured and, in most cases, (iv) the subject is acting at the center of the scene. On the other hand, the reflector-set estimation task is more complicated with respect to (i) the estimation of a larger number of reflectors in comparison with the keypoints detected by CPM approaches and (ii) the fact that the reflector patches are one-side visible. For instance, the reflectors $R_{15}$ and $R_{14}$ are both placed on the right shoulder, but on the front and the back side of the body respectively, while the right shoulder keypoint in CPM 2D poses is unique for all views.

The overall FCN method is illustrated in Figure 5. A pair of images, the colorized depth $I_{CD}^{v}$ and the corresponding 3D optical flow $I_{F}^{v}$, are given as input. A FCN simultaneously predicts a set of 2D confidence maps S of the reflector locations and a set of 2D vector fields L; the latter corresponds to the optical flow fields (OFFs) from the previous frame to the next one, encoding the temporal correlation between sequential frames. Both sets contain R=26 elements, one per reflector $R_{i}\in \left\{R_{1},\ldots ,R_{26}\right\}$, the set $S=(S_{R_{1}},S_{R_{2}},...,S_{R_{26}})$, $S_{R_{i}}\in R^{w\times h}$, where *w* and *h* are the width and height of the input images respectively, and the set $L=(L_{R_{1}},L_{R_{2}},\ldots ,L_{R_{26}})$, where $L_{R_{i}}\in R^{2\times w\times h}$. Finally, a greedy inference step is applied on the extracted confidence maps and OFFs, resulting in 2D reflector-set estimation.

#### 3.3.1. FCN Architecture

The FCN architecture, shown in Figure 6, introduces a new two-stream, two-branch, multi-stage CPM-based approach which consumes colorized depth $I_{CD}^{v}$ and 3D optical flow $I_{F}^{v}$ images. Both input streams are separately processed by a convolutional network of 10 layers (first 10 layers of VGG-19 [52]), generating two sets of feature maps $F_{D}$ and $F_{OF}$, correspondingly. Sequentially, an early stage fusion takes place, concatenating the feature maps of the two streams, $F=F_{OF}⊕F_{D}$. Let us denote *t* the stage of the network. At the first stage (t=1), the fused feature set F is given to both branches producing confidence maps, $S^{t}=\rho^{t}\left(F\right)$, and 2D vector fields, $L^{t}=\varphi^{t}\left(F\right)$, where $\rho^{t}$ and $\varphi^{t}$ denote the inference of each FCN branch. For all the subsequent T−1 stages, where *T* denotes the total number of stages, the predictions from both branches in the previous stage, along with the features set F, are fused and used to produce refined predictions based on:(1)$$\begin{array}{c} S^{t}=\rho^{t}\left(F,S^{t-1},L^{t-1}\right),t\in \left\{2,\ldots ,T\right\}\\ L^{t}=\varphi^{t}\left(F,S^{t-1},L^{t-1}\right),t\in \left\{2,\ldots ,T\right\} \end{array}$$
where the number of stages *T* is equal to 6, experimentally set by evaluating the results on the validation dataset for T=3 and T=6 stages, as proposed in [14,16], respectively. At the end of each stage, two $L_{2}$ loss functions, $L_{S}^{t}$ and $L_{L}^{t}$, between the predictions and the ground truth are applied to guide the network branches to predict confidence maps and OFFs, respectively. At stage *t*, for a 2D location p=(x,y), $p\in R^{2}$, the loss functions are given by:(2)$$\begin{array}{c} L_{S}^{t}=\sum_{r=1}^{R}\sum_{p}\left|\right|S_{r}^{t}\left(p\right)-S_{r}^{*}\left(p\right){\left|\right|}_{2}^{2}\\ L_{L}^{t}=\sum_{r=1}^{R}\sum_{p}\left|\right|L_{r}^{t}\left(p\right)-L_{r}^{*}\left(p\right){\left|\right|}_{2}^{2} \end{array}$$

In that way, the vanishing gradient problem is addressed by the intermediate supervision at each stage, replenishing the gradient periodically. The overall loss function L of the network is given by:(3)$$\begin{array}{c} L=\sum_{t=1}^{T}\left(L_{S}^{t}+L_{L}^{t}\right) \end{array}$$

#### 3.3.2. Confidence Maps and Optical Flow Fields

$L_{S}^{t}$ and $L_{L}^{t}$ are evaluated during training by generating the ground truth confidence map $S_{R_{i}}^{*}$ (Equation (4)) and vector fields $L_{R_{i}}^{*}$ (Equation (5)), respectively.

**Confidence maps:** Each confidence map is a 2D representation of the belief that a reflector occurs at each pixel location. The proposed method performs single person motion capture, therefore, a single peak should exist in each confidence map. Let $x_{R_{i},f}\in R^{2}$ be the ground truth 2D location of the reflector $R_{i}$ on the image, at frame *f*. For every 2D location $p\in R^{2}$, the ground truth value of $S_{R_{i},f}^{*}$ is given by:(4)$$\begin{array}{c} S_{R_{i},f}^{*}\left(p\right)=\exp\left(-\frac{||p-x_{R_{i},f}{\left|\right|}_{2}^{2}}{\sigma^{2}}\right) \end{array}$$
where σ controls the spread of the peak. At test time, non-maximum suppression is applied on the predicted confidence maps to localize the reflectors, assigning the confidence map peak value to each reflector prediction confidence, $E_{R_{i},f}^{S}$.

**Optical Flow Fields:** In this work, the feature representation of 2D vector fields proposed in [14], is used in a temporal manner. Preserving both 2D location and orientation information across a region, a 2D vector field for each reflector is defined by connecting the reflector 2D locations between f−1 and *f* frames. Let $x_{R_{i},f-1},x_{R_{i},f}\in R^{2}$ be the ground truth 2D locations of the reflector $R_{i}$ at frame f−1 and *f*, respectively. The ground truth value for every 2D location $p\in R^{2}$ of $L_{R_{i},f}^{*}$ is given by:(5)$$\begin{array}{c} L_{R_{i},f}^{*}\left(p\right)=\left\{\begin{array}{cc} v, & if\;p\;on\;optical\;flow\;field\\ 0, & otherwise \end{array}\right.\\ v=\frac{x_{R_{i},f}-x_{R_{i},f-1}}{\left|\right|x_{R_{i},f}-x_{R_{i},f-1}{\left|\right|}_{2}} \end{array}$$

The set of points that belong to the optical flow field includes the points within a distance threshold from the line segment between the reflector 2D locations, given by:(6)$$\begin{array}{c} 0\leq v\;\cdot \;\left(p-x_{R_{i},f-1}\right)\leq d_{R_{i}}\\ |v_{⊥}\;\cdot \;\left(p-x_{R_{i},f-1}\right)|\leq \sigma_{R_{i}} \end{array}$$
where $\sigma_{R_{i}}$ is the width of the field in pixels, $d_{R_{i}}$ is the euclidean distance of the $R_{i}$ reflector 2D locations between sequential frames in pixels, i.e., $d_{R_{i}}=\left|\right|x_{R_{i},t}-x_{R_{i},t-1}{\left|\right|}_{2}$, and $v_{⊥}$ is a vector perpendicular to v. As an example, the 2D optical flow field for the reflector $R_{13}$ is illustrated in Figure 5.

During inference, the optical flow, encoding the temporal correlations, is measured by computing the line integral over the corresponding optical flow field along the line segment connecting the candidate reflector locations between two sequential frames. Let $r_{R_{i},f}$ and $r_{R_{i},f-1}$ be the predicted locations for the reflector $R_{i}$ at the current frame *f* and the previous one f−1, correspondingly. The predicted optical flow field $L_{R_{i},f}$ is sampled along the line segment to measure the temporal correlation confidence between the predicted reflector positions in time by:(7)$$\begin{array}{c} E_{R_{i},f}^{L}=\int_{0}^{1}L_{R_{i},f}\left(p\left(u\right)\right)\;\cdot \;\frac{r_{R_{i},f}-r_{R_{i},f-1}}{\left|\right|r_{R_{i},f}-r_{R_{i},f-1}{\left|\right|}_{2}}du \end{array}$$
where p(u) interpolates the reflector positions $r_{R_{i},f}$ and $r_{R_{i},f-1}$ between sequential frames, as given by:(8)$$\begin{array}{c} p\left(u\right)=\left(1-u\right)\;\cdot \;r_{R_{i},f-1}+u\;\cdot \;r_{R_{i},f} \end{array}$$

In other words, the integral is approximated by sampling and summing uniformly spaced values of *u*.

**Greedy Inference:** Finally, a greedy inference step is introduced, taking into consideration the temporal correlations between temporally sequential 2D reflector estimates. The confidence values $E_{R_{i},f}^{S}$ and $E_{R_{i},f}^{L}$ given by the confidence maps and the optical flow fields correspondingly, are summarized in a weighted manner, in order to give the fused confidence $E_{R_{i},f}$ for each reflector estimate. In detail, the major component of $E_{R_{i},f}$ is $E_{R_{i},f}^{S}$, however, we weight the confidence $E_{R_{i},f}^{L}$ based on a $w_{R_{i},f}^{L}=\left(1-E_{R_{i},f}^{S}\right)$ factor that increases when $E_{R_{i},f}^{S}$ decreases as:(9)$$\begin{array}{c} E_{R_{i},f}=E_{R_{i},f}^{S}+w_{R_{i},f}^{L}\;\cdot \;E_{R_{i},f}^{L} \end{array}$$

In that way, when a confidence map prediction results in low confidence $E_{R_{i},f}^{S}$ estimates, the total confidence $E_{R_{i},f}$ is strongly affected by the optical flow confidence, if high. The final outcome of the 2D reflector-set estimation is given by applying non-maximum suppression on the reflector predictions based on the total confidence $E_{R_{i},f}$.

### 3.4. 3D Optical Data

#### 3.4.1. 2D-to-3D Spatial Mapping

Given the reflector-set 2D locations on the depth image $I_{D}^{v}$, a 3D spatial mapping technique is applied to precisely extract the corresponding 3D positions. Considering that the reflector locations are given exclusively when the reflectors are clearly visible, a reflector estimate is considered valid only if it belongs to a region of more than $b_{min}$ black pixels in $I_{CD}^{v}$, otherwise it is dropped. The minimum accepted size in pixels for a region was set $b_{min}=5$, since the same region size was used for the annotation of the data.

In Figure 7, two of the potential cases with respect to the reflector spatial mapping are shown. In the first case (Figure 7a), the simple and most common one, $E_{0}\in I_{CD}^{v}$ is the detected region for a reflector $R_{i}\in \left\{R_{1},\ldots ,R_{26}\right\}$. Retrieving a pixel set $P_{R_{i}}^{v}$ of the $E_{0}$ region contour in $I_{CD}^{v}$ (red pixels in Figure 7a), and mapping its points to $I_{D}^{v}$, the corresponding raw depth values of $P_{R_{i}}^{v}$ are given. Using only the non-zero depth values of $P_{R_{i}}^{v}$, the median value $d_{R_{i}}$ is considered the distance of the reflector $R_{i}$ from the sensor view *v*.

The second case is illustrated in Figure 7b, where two or more (although not usual) reflectors belong to the same region $E_{1}$. Examining the overlapping between the reflector areas, i.e., when n>1 reflectors share the same black region, the pixels of the contour are clustered in *n* clusters, based on the 2D pixel coordinates and the depth values, and then mapped to the corresponding reflectors. Subsequently, $d_{R_{i}}$ is determined for each reflector $R_{i}$ based on the clustered pixel set.

After one-to-one mapping between reflectors and regions, the central 2D point $p_{R_{i}}$ of each $E_{R_{i}}$ region is spatially mapped to 3D coordinates using the depth distance value $d_{R_{i}}$ and the intrinsic parameters of the corresponding IR-D sensor, giving the 3D position $P_{R_{i}}^{v}$ of the reflector $R_{i}$ from viewpoint *v*.

#### 3.4.2. 3D Point Sets Fusion

Using the extrinsic calibration parameters of each sensor, the extracted 3D positions are spatially aligned to a global 3D-coordinate system, as shown in Figure 8. For the reflective patches, which are one-side visible to the sensors, the same retro-reflective region is captured by all IR-D sensors and, therefore, the 3D mapping yields slightly different estimates. To this end, the 3D points $P_{R_{i}}^{v}$ for a patch reflector $R_{i}$ for all views *v* are fused to one single 3D point $P_{R_{i}}$, taking into account the confidence value $E_{R_{i}}^{v}$ of the FCN reflector estimation. The 3D point $P_{R_{i}}$ is estimated as the weighted central position of the 3D points from all views by:(10)$$\begin{array}{c} w_{R_{i}}^{v}=E_{R_{i}}^{v}/\sum_{v=1}^{N}E_{R_{i}}^{v} \end{array}\;\;\begin{array}{c} P_{R_{i}}=\frac{1}{N}\sum_{r=1}^{N}w_{R_{i}}P_{R_{i}}^{v} \end{array}$$

For the reflective straps, since different parts of the reflectors are visible to each sensor, the 3D points are estimated in different 3D locations around the part of the limb where the strap is placed on. In this case, the desired 3D point is approximately located to the center of the “circle” where these 3D points lie on, extracted by the method presented in Appendix A.

The full set of the global retro-reflector 3D positions per multi-view group frame *f*, i.e., the extracted 3D optical data, is denoted as $P_{f}$, while a total confidence value $C_{f}^{R_{i}}$ for each $R_{i}$ reflector is considered as the average value of the reflector estimation confidence $E_{R_{i}}^{v}$ for each v-view, v∈{1,….N}, $C_{f}^{R_{i}}=\frac{1}{N}\sum_{v=1}^{N}E_{R_{i}}^{v}$. To refine the quality and stability of the 3D point estimates in time, Kalman filtering [54] is applied to the extracted 3D optical data.

### 3.5. Motion Capture

The final stage of the proposed method targets at 3D motion capture based on 3D optical data. At this point, the extracted 3D optical data $P_{f}$ are mapped to a relative motion representation consisting of joint 3D positions and orientations. Similarly to [55,56], a body structure calibration technique is proposed, adapting an articulated humanoid template model to the real body structure of the subject. Subsequently, the calibrated articulated body is jointly moved by applying forward kinematics.

The proposed articulated body structure consists of 20 joints, j∈J, including $D_{DoF}=40$ DoFs. It contains $L_{i}\in L,i\in \left\{0,\ldots ,6\right\}$, hierarchical joint levels and the bones of the structure are registered to particular reflectors. To this end, a reflector subset $R_{S_{j}}\subset \left\{R_{1},\ldots ,R_{26}\right\},j\in J$, moves the body joint j∈J according to the body joint hierarchy. The correspondence between the joints and the reflectors is depicted in Figure 9a, while the reflector-to-body part mapping is illustrated in Figure 9b.

Initially, the template is coarsely scaled based on the first batch of optical data frames $P_{f}$. Then, given the 3D positions $P_{j}$, j∈J per frame, a per-bone alignment process is performed to precisely fit the body parts of the template to the subject’s real body lengths. This step is sequentially performed to the bones following the joint hierarchy levels, from $L_{0}$ to $L_{6}$. More specifically, after fitting the template to the optical data, the body root 3D position $P_{HIPS}$ with hierarchical level $L_{0}$ is given, enabling the sequential estimation of the rest of the bone lengths. Based on the assumption that the bone lengths are constant (rigid bones) and using exclusively high-confident ($C_{f}^{R_{i}}>0.6$, experimentally set) optical data $P_{f}$, a random particle generation step is applied per level in L−1 phases. More specifically, a set $G_{j}$ of G=500 particles (experimentally set) is generated around the *j*-joint location $P_{j}$ given by the spatially aligned template placement using $P_{f}$. After the particle generation, the $G_{j}$ particles relatively follow the optical data applying forward kinematics. The particle $g_{j}\in G_{j}$ that moves more rigidly between $P_{j_{l}}$ and $P_{j_{l+1}}$, is considered as the closest one to the real relative position of joint $j_{l+1}$. The objective function that estimates this particle is given by:(11)$$\begin{array}{c} D_{0}=\left|\right|P_{j_{l},0}-P_{g_{j_{l+1}},0}{\left|\right|}_{2}+\left|\right|P_{j_{l+1,0}}-P_{g_{j_{l+1}},0}{\left|\right|}_{2}\\ \underset{g_{j_{l+1}}}{arg min}D\left(g_{j_{l+1}}\right)=\frac{1}{F}\sum_{f=1}^{F}(D_{0}-\left|\right|P_{j_{l},f}-P_{g_{j_{l+1}},f}{\left|\right|}_{2}+\left|\right|P_{j_{l+1},f}-P_{g_{j_{l+1}},f}{\left|\right|}_{2}) \end{array}$$
where $D_{0}$ denotes the sum of the 3D euclidean distances at the initial frame of the (l)−(l+1) level alignment between the 3D position of the particle $g_{j_{l+1}}$ and the joints $j_{l}$ and $j_{l+1}$, as given by the latest template fitting, $P_{g_{j_{l+1}},f}$ denotes the 3D position of the particle $g_{j_{l+1}}$ at frame f∈{1,…,F}, where *F* is the total number for a window of frames used to align a body part of level l+1. In a similar fashion, the next level joints and the corresponding bones are calibrated. Angular body part movements, especially elbow and knee flexions, enable faster and more efficient convergence of the per-bone alignment process.

## 4. Evaluation Datasets

Two public datasets have been created (https://vcl.iti.gr/deepmocap/dataset) consisting of subjects wearing retro-reflectors on their bodies. These datasets are exploited for: (i) motion capture evaluation in comparison with recent MoCap methods and ground truth and (ii) 2D reflector-set estimation FCN training and testing.

### 4.1. DMC3D Dataset

The DMC3D dataset consists of multi-view depth and skeleton data as well as inertial and ground truth motion capture data. Specifically, 3 Kinect for Xbox One sensors were used to capture the depth and Kinect skeleton data along with 9 XSens MT [57] inertial measurement units (IMU) to enable the comparison between the proposed method and inertial MoCap approaches based on [44]. Furthermore, a PhaseSpace Impulse X2 [5] solution was used to capture ground truth MoCap data. PhaseSpace Impulse X2 is an optical marker-based MoCap system considered appropriate for the scope of this dataset due to the usage of active instead of passive retro-reflective markers that would have interfered with the retro-reflectors. The preparation of the DMC3D dataset required the spatio-temporal alignment of the modalities (Kinect, PhaseSpace, XSens MTs). The setup [48] used for the Kinect recordings provides spatio-temporally aligned depth and skeleton frames. For the Kinect – IMU synchronization, a global clock was used to record depth and inertial data with common timestamps. Additionally, given the timestamps and using the robust recording frequency of PhaseSpace Impulse X2 as reference clock, the spatio-temporal alignment of the ground truth data was manually achieved.

With respect to the amount and the variety of data, 10 subjects, 2 females and 8 males, wearing retro-reflectors, inertial sensors and active markers (LEDs) on the body, were recorded performing 15 physical exercises, presented in Table 1. The full dataset contains more than $80\times 10^{3}$ three-view depth and skeleton frames, the extrinsic calibration parameters and the corresponding inertial and MoCap data.

### 4.2. DMC2.5D Dataset

A second set comprising 2.5D data (DMC2.5D Dataset) was captured in order to train and test the proposed FCN. Using the recorded IR-D and MoCap data, colorized depth and 3D optical flow data pairs per view were created, as described in Section 3.2. The samples were randomly selected from 8 of the 10 subjects, excluding 2 of them to use them for the evaluation of the MoCap. More specifically, $25\times 10^{3}$ single-view pair samples were annotated with over $300\times 10^{3}$ total keypoints (i.e., reflector 2D locations of current and previous frames on the image), trying to cover a variety of poses and movements in the scene. $20\times 10^{3}$, $3\times 10^{3}$ and $2\times 10^{3}$ samples were used for training, validation and testing the FCN model, respectively. The annotation was realized by applying image processing and 3D vision techniques on the IR-D and MoCap data. In particular, applying blob detection on the IR binary image $I_{IR_{m}}^{v}$ yielded the 2D locations of the reflectors, while then, the corresponding 3D positions were estimated by applying 3D spatially mapping (Section 3.4.1). Finally, the reflectors were labeled by comparing the euclidean 3D distance per frame between the extracted 3D positions and the joint 3D positions of the ground truth data. However, the automatic labeling was erroneous in the cases that the reflector regions were merged (Figure 10) or the poses where complex. The complex pose issues occurred due to the positional offset between the reflector and the joint 3D positions, confusing the labeling process. Thus, further processing was required in order to manually refine the dataset.

## 5. Evaluation

For the evaluation of the proposed method, two types of experiments were conducted, presented and discussed in this section. The first experiment concerns the evaluation of the 2D reflector-set estimation on the DMC2.5D dataset, while, in the second one, the motion capture results are compared against robust MoCap solutions and ground truth on the DMC3D dataset. At first, FCN architectures are evaluated, highlighting the outperformance of the proposed FCN model. Accurate 2D reflector-set estimation eliminates the errors in 3D optical data extraction and, consequently, to the final motion capture outcome. Thus, afterwards, applying the proposed FCN model to the DMC3D testing dataset, 3D optical data from multi-view sequences are extracted and accordingly used for motion capture.

### 5.1. 2D Reflector-Set Estimation on DMC2.5D

#### 5.1.1. FCN Implementation

With respect to the training of the proposed FCN, data augmentation takes place randomly to increase variation of input by scaling and rotating the data. For each batch of frames being fed into the network per iteration, the transformation is consistent, thus, lower batch size results in higher variation between the iterations. Both input images are randomly scaled by $f_{s}\in \left[0.6,1.1\right]$, randomly rotated by $f_{\theta }\in \left[-40{}^{\circ},40{}^{\circ}\right]$ and flipped with binary randomness. Finally, all images are cropped to 368×368 resolution size, also setting the subject bodies at the image center. Regarding the method parameterization, the stages are equal to T=6, using stochastic gradient descent with momentum $\alpha_{m}=0.9$ and weight decay $w_{d}=5\times 10^{-4}$ to optimize the learning process. The batch size is equal to 10, while the initial learning rate is $l_{r}=2\times 10^{-5}$ and drops by $f_{g}=3.33\times 10^{-1}$ every $30\times 10^{3}$ iterations.

#### 5.1.2. Experimental Framework

The introduced FCN architecture approaches 2D-reflector-set estimation, a similar, yet different task in comparison with 2D pose estimation. Aiming to evaluate the present FCN approach and the introduced extension with respect to the temporal correlations between the reflector 2D positions from frame-to-frame, existing methods for keypoint are adapted and trained to address the present challenge. In detail, the methods by Wei et al. [16] and Cao et al. [14], included in OpenPose (https://github.com/CMU-Perceptual-Computing-Lab/openpose) library, have been adapted and trained with the DMC2.5D dataset.

For the adaptation, the body parts have been replaced by the reflectors, while it is worth noting that the Part Affinity Fields (PAFs) have been altered due to the difference of the reflector sub-set placement between the front and the back side of the body. The adapted association between the reflectors is illustrated in Figure 11. Moreover, since the proposed method has been developed for single-person motion capture, even though PAFs branch contributes to the final feature space for the confidence map prediction, its output is not taken into account for the final reflector-set estimation. Finally, a two-branch colorized depth and 3D optical flow two-stream approach similar to [14] is evaluated ([14] + 3D OF), showing remarkable results.

With respect to the evaluation metrics, the proposed FCN is evaluated on the DMC2.5D dataset measuring Average Precision (AP) per reflector and mean Average Precision (mAP) for the entire set, based on Percentage of Correct Keypoints (PCK) [58] thresholds, i.e., a prediction is considered true if it falls within a pixel region around the ground-truth keypoint. This region is defined by multiplying the width and height of the bounding box that contains the subject by a factor α that controls the relative threshold for correctness consideration.

Setting α=0.05, the validation set of the DMC2.5D dataset was used to indicate the optimum minimum confidence threshold $c_{min}$ for the highest mAP per method, aiming at fair comparison between them. $c_{min}$ corresponds to the minimum threshold of confidence for a reflector prediction to be considered as valid, i.e., $E_{R_{i},f}>c_{min}$. The results are presented in Figure 12, showing the method mAP against confidence threshold. Maximum mAP on the validation set was achieved for $c_{min}=0.4$ for all methods, thus, considered optimum for the experiments on the DMC2.5D testing set.

#### 5.1.3. Results and Discussion

Evaluating the AP results per reflector, shown in Table 2, the efficiency of the methods in reflector-set estimation is perceived. The proposed FCN method outperforms the rest of the methods for the 80.7% of the reflectors (i.e., 21 out of 26). In particular, the AP is improved for the end-reflectors, i.e., the reflectors placed on the hands and the feet ($R_{13},R_{18},R_{22},R_{26}$), and their linked reflectors, i.e., the wrists and the ankles ($R_{12},R_{17},R_{21},R_{25}$), which are placed on the body parts with the highest moving capability and, therefore, the most rapid movements. From these results, we conclude that the temporal information implicitly encoded in the proposed FCN model improves the distinction among these reflectors, while for the reflectors that the AP is slightly lower ($R_{8},R_{10},R_{13},R_{14},R_{20}$), we can assume that the predicted optical flow was not accurate or informative enough to boost the prediction confidence and, therefore, the accuracy of the estimates.

However, for all the respective methods, the prediction of the end-reflectors is weak in comparison with the rest of the reflectors. That is probably due to the fact that these reflectors are not often visible and are closely placed to their linked reflectors. To highlight this difference, mAP results are presented in Table 3 with and without the end-reflectors.

The proposed approach outperforms the competitive methods, presenting an absolute increase of mAP equal to 1.47%, 0.94% and 0.89% with end-reflectors, and 1.5%, 1.16% and 0.9% without them, in comparison with [14,16] and [14] + 3D OF, respectively. It is worth noting that the two-stream approach which takes into account the 3D optical flow ([14] + 3D OF) achieves higher mAP than [14,16], meaning that the temporal information given by the 3D optical flow stream is encoded in the feature space of the model, resulting in higher localization accuracy.

Finally, before feeding the motion capture method with the 2D reflector-set estimates, a filtering process is applied based on two fundamental considerations of the task. At first, the reflectors are detected only when visible; (i) if the region where a detector is located does not belong to a specific color (black) region of size greater than or equal to $b_{min}=5$ pixels, this estimate is discarded, and (ii) when more than one reflectors are detected on the same location (absolute distance less than 3 pixels, experimentally set), the reflectors with lower confidence are dropped. Secondly, the reflectors are unique on an image since we approach single-person MoCap; if more than one reflector estimates of the same reflector are detected, only the one with the highest confidence is considered valid.

The AP results per reflector after filtering are shown in Table 4. As shown, the results for all reflectors and for all methods are equal or greater than the corresponding results before filtering. At this experiment, the proposed FCN outperforms the rest of the methods at 14 of 26 (53.84%) of the reflectors.

The mAP results for the total set of reflectors after filtering, with and without the end-reflectors, are presented in Table 5, all showing higher accuracy than the corresponding values before filtering. The proposed method outperforms [14,16] and [14] + 3D OF by presenting an absolute increment equal to 1.25%, 0.49% and 0.37% with end-reflectors, and 1.06%, 0.47% and 0.61% without them, respectively.

In that way, the outcome of the 2D reflector-set estimation allows us to detect the 2D locations of the reflectors and, subsequently, the corresponding 3D optical data in a global coordinate system. Qualitative results of the proposed FCN outcome on sequential input data are illustrated in Figure 13, while more qualitative results have been made publicly available (https://vcl.iti.gr/deepmocap).

### 5.2. Motion Capture on DMC3D

#### 5.2.1. Experimental Framework

After the introduction of an efficient FCN model for 2D reflector-set estimation, the final MoCap outcome is evaluated. Applying the proposed qualified FCN model to multi-view sequences, the 3D optical data are extracted and fed to the MoCap proposed method. For these experiments, several sequences of approximately $6\times 10^{3}$ frames in total were selected from 2 subjects that were excluded from the dataset used to train the FCN model. Considering the ground truth data of the DCM3D dataset, i.e., the motion data captured with PhaseSpace Impulse X2 [5], the motion capture outcome is compared against the Kinect for Xbox One skeleton tracking with the highest quality, an inertial MoCap approach that fuses Kinect skeleton and inertial data from 9 IMU (Fusion) [44], and a second robust inertial method in a similar fashion as [44] (Fusion++) that fuses ground truth data for initialization and root positioning instead of Kinect skeleton tracking. It is worth noting that jerky skeleton estimates of Kinect skeleton tracking that cause highly erroneous root 3D position estimates have been excluded from the testing sequences, keeping only estimates meaningful for comparison.

Inertial MoCap methods were considered appropriate for a fair comparison due to their robust way of capturing, independent from self-occlusions. Multiple RGB-D- or 3D-based MoCap approaches were not taken into account due to the missing parts of depth and, therefore, missing 3D data on the body parts of the subject where the reflectors were placed on, resulting in unfair comparison. Finally, 3D MoCap methods from monocular RGB considered out of scope due to one-view and less-informative input, while motion capture methods from multiple RGB sources were not considered equal for comparison due to the blurry images of RGB streams on rapid movements.

Regarding the evaluation metrics, DeepMoCap is evaluated on the DMC3D dataset using Mean Average Error (MAE), Root Mean Squared Error (RMSE) and 3D PCK @ $a_{3D}=20$ cm metrics for the 3D euclidean distance between the outcome of the methods and the ground truth on 12 joints spanned by shoulders, elbows, wrists, hips, knees and ankles. In 3D PCK, an estimate is considered correct when the 3D euclidean distance is less than $a_{3D}$.

#### 5.2.2. Results and Discussion

The total results of the comparison between the MoCap methods are presented in Table 6, showing the outperformance of DeepMoCap in comparison with the rest of the methods. The total MAE, RMSE and 3D PCK for all sequences are 9.02 cm, 10.06 cm and 92.25%, respectively, showing the best results of all experimental methods. Fusion++ [44], Fusion [44] and Best Kinect [27] follow the proposed method presenting 88.75%, 85.93% and 83.37% in 3D PCK accuracy, respectively. Additionally, it is worth mentioning that the proposed method presents lower than 10 cm total mean average error (MAE).

In Table 7, the 3D PCK accuracy results are shown per exercise, giving evidence with respect to the strengths and the weaknesses of the methods based on the body movement type variations. DeepMoCap is qualified outperforming the rest of the methods at 12 of the 15 exercises in total (80%), efficiently capturing most of them. In detail, *Walking on the spot*, a simple and slow motion, is efficiently captured from all the methods. *Elbow* and *Knee flexion* exercises, which are simple rotational joint movements, are captured with high precision by all methods, especially from inertial MoCap approaches. It is worthnoting that DeepMoCap presents lower accuracy for *Side-steps* exercise than Fusion++ probably due to the hand placement on the reflectors of the hips resulting in merged reflector regions, which complicated the detection and identification of the involved reflectors. However, in more complex exercises as *Butt kicks left-right* and *Forward lunge left-right* where there are occlusions for Kinect and body stretching for inertial sensors placed on the body, DeepMoCap presents approximately 5% higher absolute 3D PCK accuracy than Fusion++, which follows. For *Jumping jacks*, which is a fast and complex exercise where all body parts are fully involved, DeepMoCap achieves 96.05% 3D PCK accuracy followed by Best Kinect [27], while inertial MoCap approaches fail to properly capture the shoulders 3D positions due to rigid body movement of the torso, showing lower accuracy in such exercises. For *Alternate side reaches* and *Kick-box kicking*, which are the most challenging exercises, the 3D PCK accuracy of the proposed method is 4.84% and 8.33% higher in comparison with the second best method (Fusion++), respectively. Furthermore, it should be noted that all exercises are captured by DeepMoCap in 3D PCK accuracy higher than 82.53%, showing low variation between different types of body movements.

In the plot presented in Figure 14, the total 3D PCK accuracy is given against $a_{3D}$ threshold values. DeepMoCap shows higher efficiency for all $a_{3D}$, showing high 3D PCK accuracy from low threshold values (e.g., 32.25% and 63.27% for $\alpha_{3D}=5$ cm and $\alpha_{3D}=10$ cm, respectively), in comparison with the next best method that presents 16.38% and 54.36%, respectively. Given the fact that joint positioning varies between different motion capture approaches resulting in the existence of a positional offset between the estimated 3D joint positions, we conclude that DeepMoCap shows high efficiency by presenting 32.25% of the estimates on average to be closer than 5 cm from ground truth.

In Table 8, the euclidean MAE and RMSE are presented per joint. It can be observed that the proposed method has the lowest errors for 9 of 12 (75%) joints for MAE and RMSE. For *Shoulders* and *Right Elbow*, Fusion++ [44] shows slightly better results than DeepMoCap probably due to better skeleton structure positioning. The lower body joints (hips, knees and feet) are captured presenting 6.05 cm and 7.08 cm mean average and root mean squared errors, respectively.

Qualitative results depicting the 3D outcome of the proposed approach are presented in Figure 15. In particular, the multi-view input overlayed with the reflector-set estimates and the corresponding 3D motion capture along with optical data results are illustrated. As shown, DeepMoCap approaches motion capture similarly to the way traditional optical MoCap solutions work, in a more flexible and low-cost manner though. More qualitative results are publicly available (https://vcl.iti.gr/deepmocap).

### 5.3. Performance Analysis

The runtime performance analysis was conducted by measuring the total time required to capture the motion data from the testing sequences. For approximately $6\times 10^{3}$ three-view pairs of colorized depth and 3D optical flow frames, thus $18\times 10^{3}$ single-view pairs, raw data pre-processing lasted 1796 s (∼100 ms per sample), while FCN model prediction required 3136 s (∼174 ms per sample). Thus, the proposed method achieves 2D reflector-set estimation at approximately 6 frames per second with ∼100 ms latency, while the motion tracking from optical data is real-time, requiring less than 10 ms. With respect to the input, the original frame size is 424×512, re-sized to 368×444 during testing to fit in GPU memory. Thus, DeepMoCap performs motion capture at approximately 2 Hz for 3-view input of 368×444, while the performance complexity against number of views, i.e., input image pairs, is O(n). The runtime analysis was performed on a desktop machine equipped with one NVIDIA GeForce Titan X GPU. Code (https://github.com/tofis/deepmocap) and dataset tools of the proposed method are publicly available to encourage further research in the area.

## 6. Conclusions

In the present work, a deep marker-based optical motion capture method is introduced, using multiple IR-D sensors and retro-reflectors. DeepMoCap constitutes a robust, fast and flexible approach that automatically extracts labeled 3D optical data and performs immediate motion capture without the need for post-processing. For this purpose, a novel two-stream, multi-stage CPM-based FCN is proposed that introduces a non-parametric representation to encode the temporal correlation among pairs of colorized depthmaps and 3D optical flow frames, resulting in retro-reflector 2D localization and labeling. This step enables the 3D optical data extraction from multiple spatio-temporally aligned IR-D sensors and, subsequently, motion capture. For research and evaluation purposes, two new datasets were created and made publicly available. The proposed method was evaluated with respect to the 2D reflector-set estimation and the motion capture accuracy on these datasets, outperforming recent and robust methods in both tasks. Taking into consideration this comparative evaluation, we conclude that the joint usage of traditional marker-based optical MoCap rationale and recent deep learning advancements in conjunction with 2.5D and 3D vision techniques can significantly contribute to the MoCap field, introducing a new way of approaching the task.

With respect to the limitations, the side-view capturing and the highly complex body poses that occlude or merge reflectors on the image views constitute the main barriers. These limitations can be eliminated by increasing the number of IR-D sensors around the subject, however, increasing the cost and complexity of the method. Next steps of this research would include the study of recent deep learning approaches in 3D pose recovery and motion capture, investigating key features that will allow us to address main MoCap challenges such as real-time performance, efficient multi-person capturing, in outdoor environments, with more degrees of freedom of the body to be captured and higher accuracy.

## Figures and Tables

**Figure 1 sensors-19-00282-f001:**
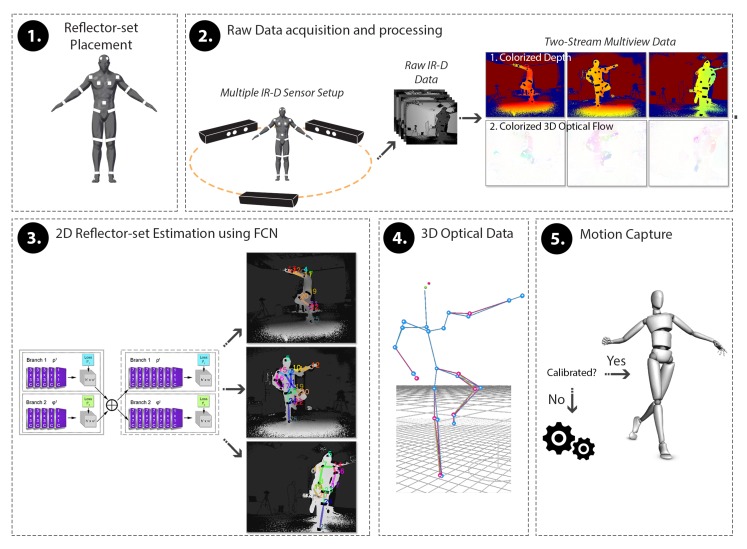
DeepMoCap pipeline overview. After the placement of the reflectors on the subject’s body (1), spatio-temporally aligned IR-D streams are acquired and processed (2) to feed the FCN with colorized depth and 3D optical flow images (3). The FCN outcome, i.e., the multi-view 2D reflector-set estimates, is fused to extract the 3D optical data (4) and, finally, yield the subject’s 3D pose for motion capture (5).

**Figure 2 sensors-19-00282-f002:**
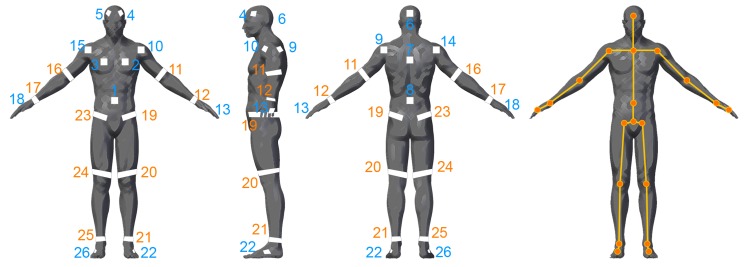
Proposed reflector-set placement. Reflective straps (orange) and patches (blue) placement on the subject’s body.

**Figure 3 sensors-19-00282-f003:**
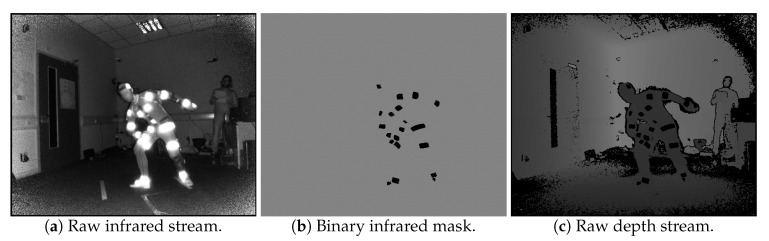
Raw depth and infrared data.

**Figure 4 sensors-19-00282-f004:**
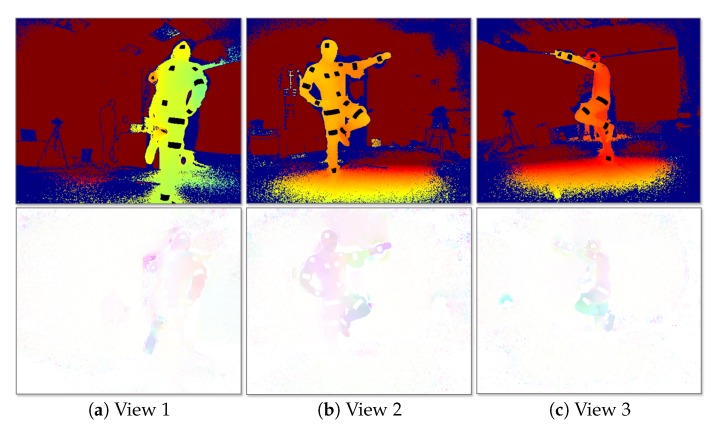
Multiview two-stream input. (**Up**) Colorized depth, mask subtracted. (**Down**) Colorized 3D optical flow.

**Figure 5 sensors-19-00282-f005:**
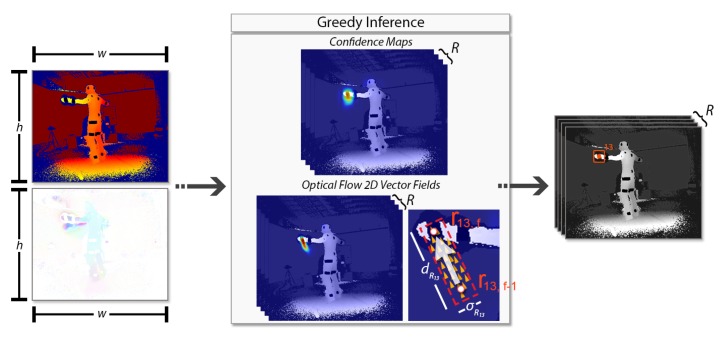
Overall 2D reflector-set estimation from confidence maps and optical flow 2D vector fields. The 2D location for $R_{13}$ reflector is estimated taking into account its predicted heat map and optical flow between the predictions $r_{13,f-1}$ and $r_{13,f}$.

**Figure 6 sensors-19-00282-f006:**
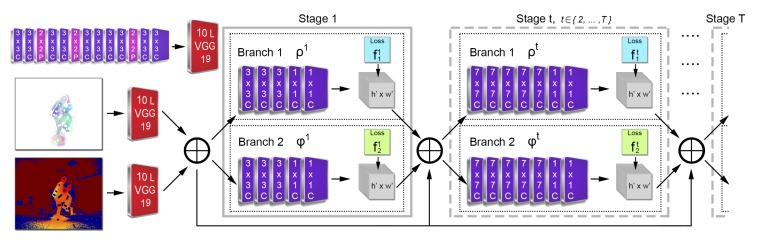
DeepMoCap two-stream, multi-stage FCN architecture. The outcome of each stage t∈{2,…,T} and the feature set **F** are fused and given as input to the next stage.

**Figure 7 sensors-19-00282-f007:**
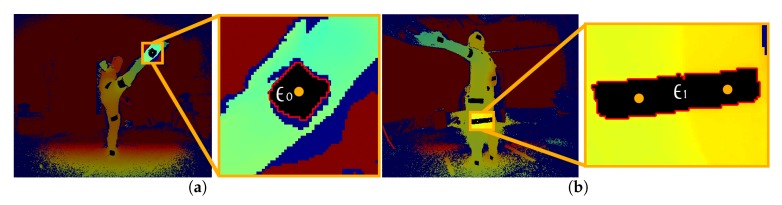
Contour detection (red pixels) of the reflector regions for depth estimation and 3D mapping. (**a**) Contour of single reflector region, $E_{0}$. (**b**) Contour of multi-reflector region, $E_{1}$.

**Figure 8 sensors-19-00282-f008:**
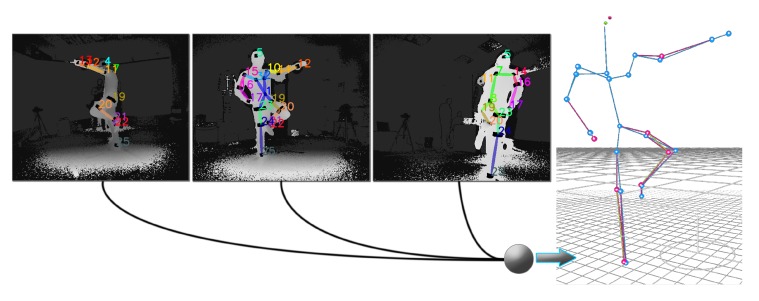
Multiple 2D reflector-set estimates (**left**) are spatially mapped in a global 3D Cartesian coordinate system, resulting in 3D optical data extraction (**right**).

**Figure 9 sensors-19-00282-f009:**
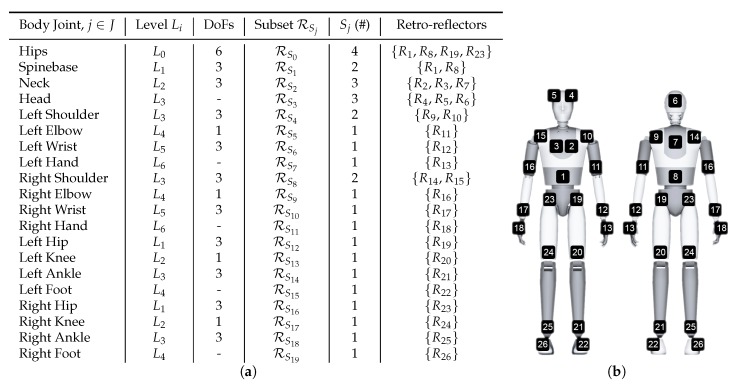
(**a**) Template model joints, hierarchical level, DoFs and correspondence to the reflector subsets. (**b**) Reflector mapping to body structure body parts.

**Figure 10 sensors-19-00282-f010:**
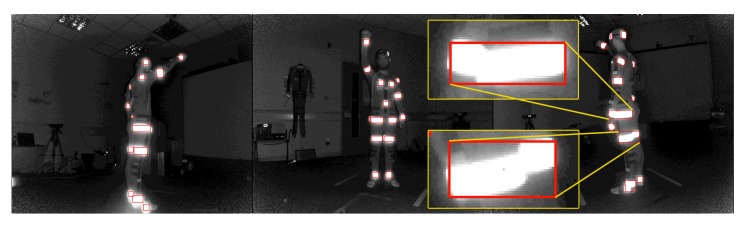
Reflector annotation using blob tracking on IR binary mask, visualized though on $I_{IR}^{v}$ IR data for the sake of better understanding. Erroneous estimates occur when reflector regions are merged.

**Figure 11 sensors-19-00282-f011:**
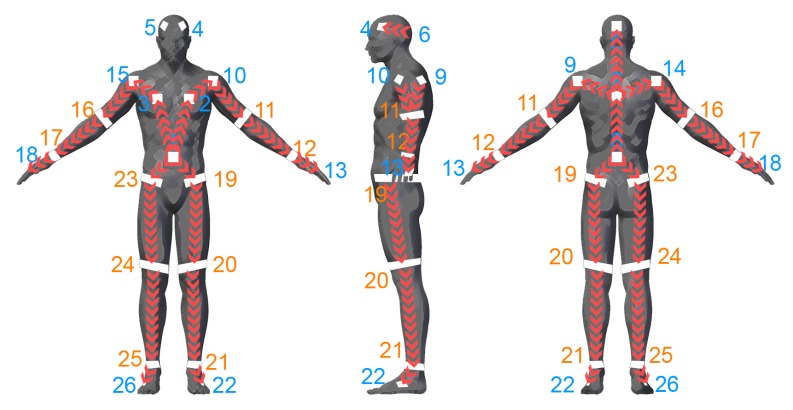
The red arrows illustrate the directional associations between the reflectors to adapt the Part Affinity Fields, as proposed in [14]. The orange and blue colored labels indicate the reflective straps and patches, respectively.

**Figure 12 sensors-19-00282-f012:**
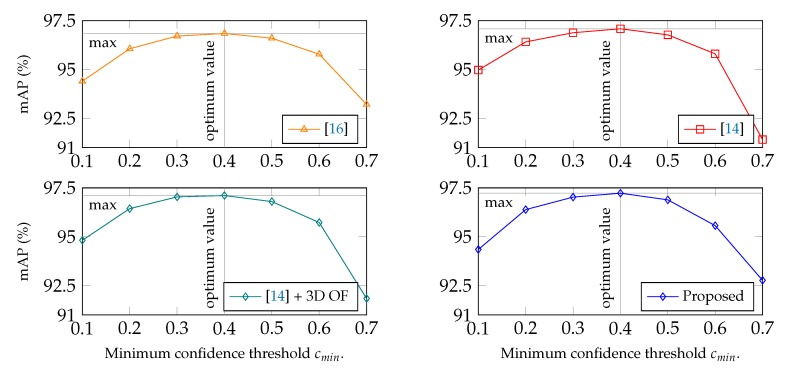
Mean Average Precision based on Percentage of Correct Keypoints thresholds (a=0.05) against confidence threshold, $mAP(c_{min})$.

**Figure 13 sensors-19-00282-f013:**
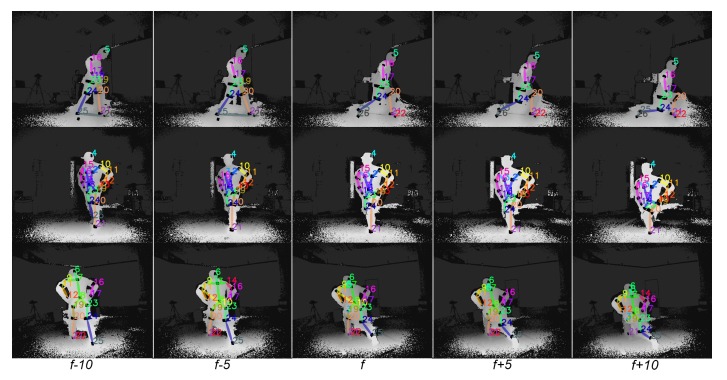
Visualization of the proposed FCN outcome overlayed on sequential multi-view depth frames. Five multi-view sequential frames, from frame f−10 to f+10 with frame step equal to 5, are horizontally presented.

**Figure 14 sensors-19-00282-f014:**
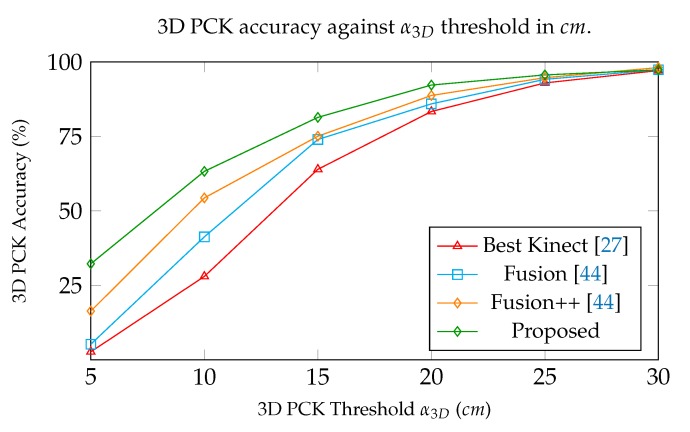
Comparative evaluation of the motion capture methods using total 3D PCK results in different $\alpha_{3D}$ threshold values in cm.

**Figure 15 sensors-19-00282-f015:**
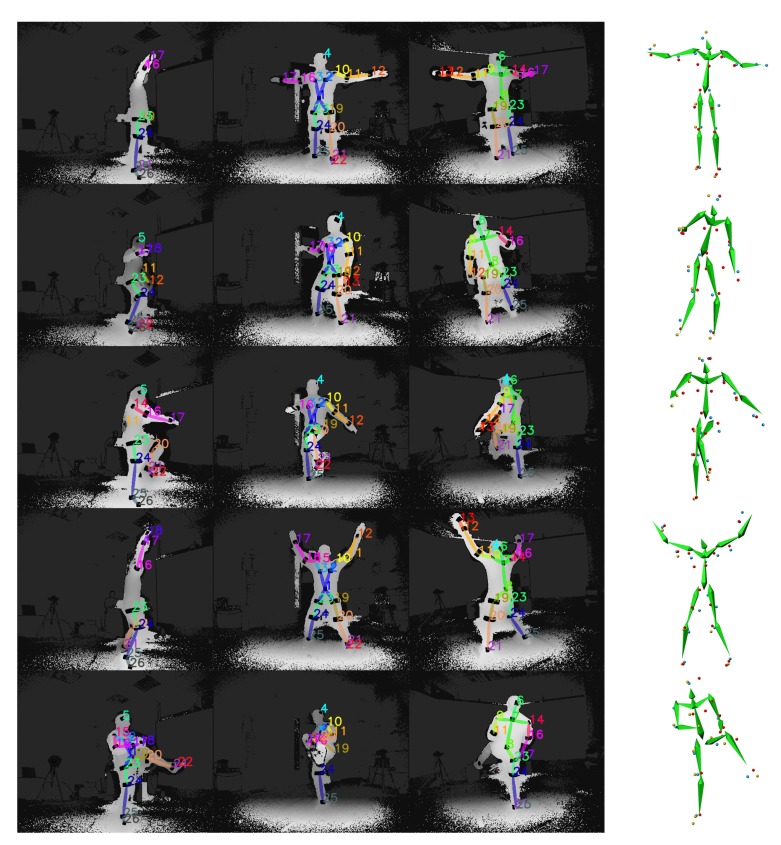
Five samples of the method results are illustrated in rows. At the left side of the figure, the multi-view input along with the FCN reflector-set estimates are presented, while, at the right side, their corresponding 3D optical and motion capture outcomes are shown.

**Table 1 sensors-19-00282-t001:** Data captured per subject in the DMC3D dataset.

Physical Exercise	# of Repetitions	# of Frames	Type
Walking on the spot	10–20	200–300	Free
Single arm raise	10–20	300–500	Bilateral
Elbow flexion	10–20	300–500	Bilateral
Knee flexion	10–20	300–500	Bilateral
Closing arms above head	6–12	200–300	Free
Side steps	6–12	300–500	Bilateral
Jumping jack	6–12	200–300	Free
Butt kicks left-right	6–12	300–500	Bilateral
Forward lunge left-right	4–10	300–500	Bilateral
Classic squat	6–12	200–300	Free
Side step + knee-elbow	6–12	300–500	Bilateral
Side reaches	6–12	300–500	Bilateral
Side jumps	6–12	300–500	Bilateral
Alternate side reaches	6–12	300–500	Bilateral
Kick-box kicking	2–6	200–300	Free

**Table 2 sensors-19-00282-t002:** AP for PCK with α=0.05, for each of the 26 reflectors. The proposed FCN method outperforms the rest of the methods for the 80.7% of the reflectors (i.e., 21 out of 26).

**%**	**R01**	**R02**	**R03**	**R04**	**R05**	**R06**	**R07**	**R08**	**R09**	**R10**	**R11**	**R12**	**R13**
[16]	96.81	96.11	99.22	95.06	90.98	85.26	98.78	**99.76**	95.25	94.70	96.99	93.33	85.92
[14]	96.95	95.36	98.77	96.69	91.08	85.26	98.78	99.51	96.00	95.89	97.15	93.79	87.64
[14] + 3D OF	96.81	96.61	99.45	94.85	89.98	85.53	98.78	99.45	96.00	**96.27**	97.25	93.49	**87.66**
Proposed	**98.10**	**97.31**	**99.48**	**97.35**	**91.36**	**86.20**	**99.00**	98.27	**96.64**	95.32	**97.81**	**95.19**	87.13
**%**	**R14**	**R15**	**R16**	**R17**	**R18**	**R19**	**R20**	**R21**	**R22**	**R23**	**R24**	**R25**	**R26**
[16]	83.42	93.21	98.24	96.25	75.23	98.39	98.38	94.52	74.28	94.88	99.30	97.18	64.73
[14]	84.58	92.88	98.66	95.95	76.19	98.39	**98.44**	94.88	74.06	96.99	99.52	97.87	71.18
[14] + 3D OF	**86.33**	94.11	98.44	95.92	72.29	98.40	98.42	95.14	78.68	96.53	99.21	97.76	70.56
Proposed	85.61	**94.79**	**98.81**	**97.50**	**77.17**	**99.30**	98.18	**96.61**	**79.23**	**97.93**	**100.0**	**98.73**	**73.96**

**Table 3 sensors-19-00282-t003:** mAP for PCK with α=0.05, with and without end-reflectors.

Method	Total	Total (without End-Reflectors)
[16]	92.16%	95.27%
[14]	92.79%	95.61%
[14] + 3D OF	92.84%	95.67%
Proposed	**93.73%**	**96.77%**

**Table 4 sensors-19-00282-t004:** AP for PCK with α=0.05, for each of the 26 reflectors, after filtering.

**%**	**R01**	**R02**	**R03**	**R04**	**R05**	**R06**	**R07**	**R08**	**R09**	**R10**	**R11**	**R12**	**R13**
[16]	96.95	96.39	99.45	97.30	91.74	86.81	100.0	**100.0**	96.62	**97.46**	98.15	94.16	90.50
[14]	96.95	96.19	98.77	98.31	**93.98**	86.81	100.0	99.76	96.87	**97.46**	98.55	94.64	90.79
[14] + 3D OF	96.81	96.88	99.45	97.30	91.72	86.81	100.0	99.70	96.87	**97.46**	98.55	94.05	**91.67**
Proposed	**98.52**	**98.61**	**99.81**	**98.65**	92.58	**87.35**	100.0	**100.0**	**98.90**	96.59	**100.0**	**95.50**	89.60
**%**	**R14**	**R15**	**R16**	**R17**	**R18**	**R19**	**R20**	**R21**	**R22**	**R23**	**R24**	**R25**	**R26**
[16]	83.78	95.93	99.03	97.19	78.81	98.60	99.41	96.59	74.08	97.59	99.40	98.36	68.64
[14]	**90.19**	96.40	**99.32**	97.17	**82.82**	98.60	99.37	**97.25**	75.66	**99.09**	99.61	98.60	69.52
[14] + 3D OF	**90.19**	**96.75**	99.14	98.30	78.87	98.50	**99.50**	96.83	**81.69**	98.25	99.30	98.52	72.68
Proposed	88.90	95.10	99.13	**97.82**	78.20	**99.63**	98.50	96.93	79.49	98.25	**100.0**	**99.05**	**78.21**

**Table 5 sensors-19-00282-t005:** mAP for PCK with α=0.05, with and without end-reflectors, after filtering.

Method	Total	Total (without End-Reflectors)
[16]	93.57%	96.41%
[14]	94.33%	97.00%
[14] + 3D OF	94.45%	96.86%
Proposed	**94.82%**	**97.47%**

**Table 6 sensors-19-00282-t006:** Comparative evaluation of the motion capture results of the respective methods, presenting total MAE, RMSE and 3D PCK ($\alpha_{3D}=20$ cm) metrics.

Method	MAE (cm)	RMSE (cm)	3D PCK (a=20 cm) [58]
Best Kinect [27]	15.35	16.06	82.03%
Fusion [44]	12.31	12.91	85.93%
Fusion++ [44]	10.66	11.30	88.75%
Proposed	**9.02**	**10.06**	**92.25%**

**Table 7 sensors-19-00282-t007:** Comparative evaluation per exercise using 3D PCK, $\alpha_{3D}=20$ cm metric.

Exercise	Best Kinect [27]	Fusion [44]	Fusion++ [44]	Proposed
Walking on the spot	96.60%	**100.00%**	97.54%	**100.00%**
Single arm raise	93.57%	96.19%	97.38%	**100.00%**
Elbow flexion	91.12%	**100.00%**	**100.00%**	97.40%
Knee flexion	88.36%	94.11%	**100.00%**	98.80%
Closing arms above head	82.48%	80.08%	83.33%	**88.62%**
Side steps	85.00%	88.33%	**93.33%**	87.50%
Jumping jack	95.48%	84.18%	87.57%	**96.05%**
Butt kicks left-right	81.87%	80.99%	86.26%	**90.94%**
Forward lunge left-right	57.31%	87.93%	86.05%	**92.01%**
Classic squat	59.60%	78.67%	83.05%	**90.40%**
Side step + knee-elbow	77.78%	80.25%	81.94%	**89.81%**
Side reaches	89.24%	84.55%	87.88%	**91.52%**
Side jumps	90.00%	89.31%	92.78%	**93.47%**
Alternate side reaches	68.01%	74.19%	77.69%	**82.53%**
Kick-box kicking	74.07%	70.14%	76.39%	**84.72%**

**Table 8 sensors-19-00282-t008:** Experimental results of the respective motion capture methods using (MAE) and (RMSE) metrics per joint (in cm).

Joint	Best Kinect [27]		Fusion [44]		Fusion++ [44]		Proposed	
	MAE	RMSE	MAE	RMSE	MAE	RMSE	MAE	RMSE
Left Shoulder	12.83	13.25	8.16	**8.37**	**7.89**	8.53	11.41	12.63
Right Shoulder	15.59	15.90	9.71	10.28	**8.62**	**9.19**	11.09	11.76
Left Elbow	16.04	17.45	16.46	17.17	15.90	16.60	**13.25**	**14.84**
Right Elbow	19.37	19.67	11.61	12.61	**10.88**	**11.68**	12.25	13.36
Left Hand	16.01	17.95	14.52	15.87	13.53	14.44	**11.94**	**12.58**
Right Hand	21.24	21.55	13.10	14.24	12.42	13.29	**12.04**	**13.05**
Left Hip	8.63	8.82	9.99	10.20	6.33	6.45	**4.18**	**4.69**
Right Hip	10.89	11.16	10.59	10.81	5.94	6.11	**4.53**	**4.99**
Left Knee	10.79	11.73	12.55	13.10	9.97	10.45	**5.12**	**5.82**
Right Knee	15.13	15.99	12.17	12.64	8.92	9.45	**7.24**	**8.16**
Left Foot	17.74	18.34	16.35	17.11	15.57	16.40	**7.00**	**8.82**
Right Foot	19.91	20.86	12.48	12.53	11.93	12.97	**8.24**	**10.00**

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
