# Peer review of "DeepMoCap: Deep Optical Motion Capture Using Multiple Depth Sensors and Retro-Reflectors"

_sensors, 2019, doi:10.3390/s19020282_

Round 1

Reviewer 1 Report

The paper at hand describes a new method for motion capturing based on multiple depth and rgb cameras. In general, the paper is well written and good to follow. I only have few questions and remarks that should be resolved in a minor revision, before the paper is ready for publication.

-          With the set of markers and the described approach, is the method sensitive to marker placement? How do changes in marker placement (bad placed markers) influence the result? The authors state that a high precision is not required, but this statement should be supported by some numbers.

-          Is there any calibration procedure needed for the camera placement? Or can I place the cameras as I want, and the system can work with the data directly?

-          Related work: Inertial based capturing was improved by estimating ground contacts (see:  Motion Reconstruction Using Very Few Accelerometers and Ground Contacts; Qaiser Riaz et al.; Graphical Models (Apr. 2015))

-          It is not clear what data is taken as ground truth (gold standard) in the experimental section. When comparing various methods using the MAE, RMSE, and 3dPCK measures, what is the basis for the computation?

Author Response

Dear Sir/Madame,

Thank you very much for reviewing our manuscript providing productive feedback with your comments. Please, find below the respective responses.

Point 1: With the set of markers and the described approach, is the method sensitive to marker placement? How do changes in marker placement (bad placed markers) influence the result? The authors state that a high precision is not required, but this statement should be supported by some numbers. 

Response 1: The method is not too sensitive to marker placement since the straps give the central point of the point which lies on the line of the limb bone axis and, in general, these central points as well as the patches relative 3D positions are defined during the body structure calibration procedure and, subsequently used to move the humanoid template. It is worth noting that the training of the method has been conducted using data from 8 subjects with different body structures, therefore slightly different reflector-set placement, and the motion capture has been evaluated on subjects that were not included in the FCN training set. However, the marker placement should follow the rules below:

-          The size of the reflectors should be close to the size of the reflectors used to train the method (patch edge and strap width of the markers used in the datasets is equal to 2 inches which is the standard width of the reflective tape used for Safety Vests and protecting clothing in general). Otherwise, the predictions of the FCN model may be affected. For this purpose, a reference has been added to the manuscript to indicate the reflective tape type used to create the reflectors.

-          The reflector-set consists of patches and straps and each body part requires the placement of a specific reflector type as shown in Figure 2. If this rule is not followed, the method will fail (e.g., use of a long strap on the chest or the head instead of patches, reflective strips that do not form a strap (band) and there are reflection gaps on the infrared data, etc.).

-          The reflectors are required to be placed on the body part following the proposed placement, otherwise the body structure won’t be moved correctly.

-          The placement is not required to be too precise (e.g., R20 and R24 could be placed 5-15 cm above the knee), however, it should be placed to the area shown in Figure 2, not to a different location (e.g., close to another reflector). Otherwise, the predictions of the FCN model will be affected and probably fail. 

Clarifications have been included in Section 3.1. Reflector-set placement to the revised manuscript.

Point 2: Is there any calibration procedure needed for the camera placement? Or can I place the cameras as I want, and the system can work with the data directly?

Response 2: For the spatiotemporal alignment between the sensors, the multi-Kinect calibration and synchronization techniques proposed by Alexiadis et al. [48] take place before the recording of a sequence. The use of the multi-Kinect capturing system by [48] that provides spatio-temporally aligned data, is reported in Section 4.1. DMC3D Dataset. 

The high accuracy of the 3D optical data extraction was one of the requirements with the highest priority, thus, an automatic calibration procedure based on the extracted marker 3D positions has not been investigated in the present work, however, it constitutes one of the next steps for the extension of the method. 

Point 3: Related work: Inertial based capturing was improved by estimating ground contacts (see: Motion Reconstruction Using Very Few Accelerometers and Ground Contacts; Qaiser Riaz et al.; Graphical Models (Apr. 2015))

Response 3: Thank you for indicating this work, it has been included in the related work and the references of the revised manuscript.

Point 4: It is not clear what data is taken as ground truth (gold standard) in the experimental section. When comparing various methods using the MAE, RMSE, and 3dPCK measures, what is the basis for the computation?

Response 4: The data used as ground truth for the evaluation of the motion capture approach is captured using PhaseSpace Impulse X2 (http://www.phasespace.com/impulse-motion-capture.html), a marker-based optical motion capture solution that uses active markers (leds) instead of reflective markers in order to avoid interference with the DeepMoCap retro-reflectors. The usage of this data as ground truth is reported in the description of the DMC3D dataset (Section 4.1. DMC3D Dataset).

Using the motion data captured with PhaseSpace and after spatiotemporally aligning it with the multi-IRD capturing system, the 3D Euclidean distances per joint per frame were calculated to give the error of the estimates with respect to ground truth. This methodology is described in Section 5.2.1. Experimental Framework.

 In order to facilitate the reader understanding, the use of PhaseSpace as ground truth has been reported in Section 5.2.1 as well.

Reviewer 2 Report

The submission describes an excellent combination of deep learning and depth sensor data for motion capture applications with multiple depth cameras. The open-data availability for further research also enhances the research and forms a good base for future research directions.

Overall the submission is well-written, a few small changes:

Line 170: utilizing of-the-shelf...- change to off-the-shelf

Line 428: data augmentation takes randomly place - takes place randomly 

The extension of the research work to the availability of online data for other researchers to experiment with should provide lasting value to re-evaluate and improve upon the methods presented in the paper. Currently the Github link does not provide direct access to the code as described on line 585, but requires emailing to the repository owner. If the paper states that code is publicly available, it should be made available as stated without further action by researchers, or the link in the paper should state that the code may be made available after contact with the repository owner.

Author Response

Dear Sir/Madame, 

Thank you very much for reviewing our manuscript providing productive feedback with your comments. Please, find below the respective responses.

Point 1: Overall the submission is well-written, a few small changes: 

Line 170: utilizing of-the-shelf...- change to off-the-shelf 

Line 428: data augmentation takes randomly place - takes place randomly 

Response 1: Thank you for highlighting these changes, they have been corrected to the revised manuscript.

Point 2: The extension of the research work to the availability of online data for other researchers to experiment with should provide lasting value to re-evaluate and improve upon the methods presented in the paper. Currently the GitHub link does not provide direct access to the code as described on line 585, but requires emailing to the repository owner. If the paper states that code is publicly available, it should be made available as stated without further action by researchers, or the link in the paper should state that the code may be made available after contact with the repository owner. 

Response 2: Thank you for the indication. The repository has been updated including a public link for direct access to the FCN model, since the file is larger than 100 mb, which is the file size limit for GitHub. Moreover, in the testing folder, a full test has been added allowing for 2D reflector-set estimation from colorized depth and 3D optical flow sample pairs using the FCN model and, subsequently, 3D spatial mapping using the intrinsic and extrinsic IR-D camera parameters. Finally, the table presenting the overall results of the methods has been included to the corresponding testing sections of the repository.
